# PAM50 Intrinsic Subtype Profiles in Primary and Metastatic Breast Cancer Show a Significant Shift toward More Aggressive Subtypes with Prognostic Implications

**DOI:** 10.3390/cancers13071592

**Published:** 2021-03-30

**Authors:** Charlotte Levin Tykjær Jørgensen, Anna-Maria Larsson, Carina Forsare, Kristina Aaltonen, Sara Jansson, Rachel Bradshaw, Pär-Ola Bendahl, Lisa Rydén

**Affiliations:** 1Division of Oncology, Department of Clinical Sciences Lund, Lund University, SE-223 81 Lund, Sweden; charlotte.levin.tykjaer.joergensen@regionh.dk (C.L.T.J.); anna-maria.larsson@med.lu.se (A.-M.L.); carina.forsare@med.lu.se (C.F.); sara.jansson@med.lu.se (S.J.); par-ola.bendahl@med.lu.se (P.-O.B.); 2Department of Hematology, Oncology and Radiation Physics, Skåne University Hospital, SE-221 85 Lund, Sweden; 3Division of Translational Cancer Research, Department of Laboratory Medicine, Lund University, SE-223 81 Lund, Sweden; kristina.aaltonen@med.lu.se; 4NanoString Technologies, Seattle, WA 98109, USA; rbradshaw@nanostring.com; 5Division of Surgery and Division of Oncology, Department of Clinical Sciences, Lund University, SE-223 81 Lund, Sweden; 6Department of Surgery, Skåne University Hospital, SE-205 02 Malmö, Sweden

**Keywords:** metastatic breast cancer, PAM50 breast cancer intrinsic subtype, tumor progression, subtype shift, tumor heterogeneity

## Abstract

**Simple Summary:**

The majority of breast cancer deaths are caused by the spread of the disease to distant locations. The biological processes and molecular characteristics that eventually transform breast cancer into a life-threatening metastatic disease are not fully understood. The molecular subtyping of breast cancer into four tumor subtypes—namely luminal A, luminal B, human epidermal growth factor receptor 2-enriched, and basal-like subtypes—has been implemented for therapeutic guidance in patients with early breast cancer. It is not settled whether molecular subtypes in metastatic tissue can guide the choice of systemic therapy and how these subtypes may change throughout tumor progression. In this study, breast cancer subtypes at different stages of the disease were investigated, and we found changes to more unfavorable subtypes to be common throughout the progression of the disease. These findings suggests that molecular subtyping in metastatic disease could add important prognostic and predictive information to complement information from the primary tumor.

**Abstract:**

Background: PAM50 breast cancer intrinsic subtyping adds prognostic information in early breast cancer; however, the role in metastatic disease is unclear. We aimed to identify PAM50 subtypes in primary tumors (PTs) and metastases to outline subtype changes and their prognostic role. Methods: RNA was isolated from PTs, lymph node metastases (LNMs), and distant metastases (DMs) in metastatic breast cancer patients (*n* = 140) included in a prospective study (NCT01322893). Gene expression analyses were performed using the Breast Cancer 360 (BC360) assay from Nano-String. The subtype shifts were evaluated using McNemar and symmetry tests, and clinical outcomes were evaluated with log-rank tests and Cox regression. Results: The PAM50 subtype changed in 25/59 of paired samples between PTs and LNMs (*P_symmetry_* = 0.002), in 31/61 between PTs and DMs (*P_symmetry_* < 0.001), and in 16/38 between LNMs and DMs (*P_symmetry_* = 0.004). Shifts toward subtypes with worse outcomes were the most common. Patients with shifts from the luminal PT to non-luminal DM subtypes had worse progression-free survival compared to patients with a stable subtype (hazard ratio (HR): 2.3; 95% confidence interval (CI): 1.14–4.68, *p* = 0.02). Conclusion: Strong evidence of PAM50 subtype shifts toward unfavorable subtypes were seen between PTs and metastatic samples. For patients with a shift in subtype from luminal PT to non-luminal DM, a worse prognosis was noted.

## 1. Introduction

The majority of breast cancer deaths can be attributed to metastasis [1]. Thus far, the biological processes and molecular characteristics of breast cancer’s progression into life-threatening disease are still incompletely described. As personalized cancer treatment is based on tumor molecular characterization, tumor heterogeneity within the same patient causes great challenges in clinical practice [2,3,4]. A number of papers have reported a discrepancy in the status of routinely used biomarkers when applied to differentiate between the primary tumor and corresponding metastases in a relatively large proportion of metastatic breast cancer (MBC) patients [5,6,7]. The routine assessment of biomarker status using core needle biopsies from metastases are therefore recommended to improve the guidance for selecting systemic therapy in the context of metastasis [8].

Throughout tumor progression, little is known regarding changes in the four major molecular subtypes of breast cancer, namely the luminal A, luminal B, human epidermal growth factor receptor 2-enriched (HER2-E), and basal-like subtypes [9,10,11,12,13,14,15,16,17]. Based on relatively few studies, shifts in the breast cancer molecular subtypes in the range of 11–55% have been recognized when comparing patient-matched primary and metastatic tumors, with the majority of these changes being from the luminal A to non-luminal A subtypes. Differences between primary and recurrent disease could have an impact on the prognosis and, hence, have implications for the choice of systemic therapy. However, an unambiguous association has not been established between shifts in molecular traits and survival outcomes [5,9,10,13,17]. The previous assessments of change in tumor inherence were based on the analysis of archival tissues and mainly presented the surrogate subtypes [5,9,10,17], although one of the studies presented data on the intrinsic molecular subtypes [13].

In the present study, we aimed to study changes in the intrinsic subtypes during tumor progression in breast cancer patients by analyzing the molecular status of tumor samples at different stages of the disease. The main objective was to analyze tumor specimens from a prospective observational trial of MBC patients with long-term follow-up data to determine the concordance of PAM50 breast cancer intrinsic subtypes in matched samples of primary tumors (PTs), synchronous lymph node metastases (LNMs), and distant metastases (DMs). A secondary objective was to evaluate the effect of subtype instability on prognosis and to assess the prognostic value of the subtype classification of metastases.

## 2. Materials and Methods

### 2.1. Patients and Tissue Samples

The design of the current study was prospective–retrospective [18] and used a patient cohort enrolled in a prospective monitoring trial (circulating tumor cell (CTC)-MBC trial; Clinical Trials.gov NCT01322893) to evaluate the prognostic value of serial enumeration of CTCs in women with newly diagnosed MBC scheduled for first-line systemic therapy [19]. In brief, the inclusion criteria were a diagnosis of MBC, age ≥18 years, an Eastern Cooperative Oncology Group (ECOG) performance status score of 0–2, and a predicted life expectancy of >2 months.

Patients were excluded if they were unable to understand the study information, if they had received prior systemic treatment for metastatic disease, or if they had been diagnosed with any other malignant disease within the previous five years. Informed consent was obtained from all patients included in the study. The Lund University Ethics Committee (LU 2010/135) approved the study prior to its initiation. Of the 168 patients initially included in the trial, eight patients were excluded because they did not fulfill the inclusion criteria and four patients were excluded because they only had loco-regional recurrence. Based on pathology reports of the remaining 156 patients, formalin-fixed, paraffin-embedded (FFPE) samples of PTs, synchronous LNMs, and DMs were collected. In cases where multiple DMs were available, the sample from the first metastatic site was used in analysis (Figure 1).

### 2.2. Macrodissection of Tumor Tissue and RNA Isolation

Sections from collected FFPE tumor tissues stained with hematoxylin and eosin were reviewed, and areas with representative invasive breast carcinoma were outlined on each slide. In doubtful cases, a pathologist (Anna Ehinger) performed a second assessment. Based on the size of the tumor surface area, a total of 1–8 tissue sections, 10 μm thick, were cut and macrodissected to remove the surrounding normal tissue outside the outlined area.

Total RNA was extracted using the AllPrep DNA/RNA FFPE kit (Qiagen Cat: 80234, Hilden, Germany) in accordance with the instructions provided by the manufacturer. RNA yield and purity were assessed using a NanoDrop ND-8000 Spectrophotometer (NanoDrop Technologies, Rockland, DE, USA). RNA quality control (QC) was performed using an Agilent 2100 Bioanalyzer Instrument (Agilent Technologies, Santa Clara, CA, USA). A subset of samples was analyzed with the Bioanalyzer as part of the QC of the RNA (mean RNA integrity number (RIN)-value was 2.3 (1.3–2.8). NanoString used multiple QC cut-offs to determine if a sample was PASS, FAIL, or BORDERLINE, which indicated if the RNA quality was high enough to produce reliable subtypes and that the NanoString assay should perform as expected. Due to these post-collection QC steps, NanoString did not need to assess RNA quality before the assay. Most samples (90%) passed the predefined quality criteria set by NanoString, and only 17 patients were excluded due to analytical failure (Figure 1).

### 2.3. Gene Expression Assay

Gene expression was measured using the NanoString Breast Cancer 360 assay (BC360^TM^) on an NanoString nCounter^®^ SPRINT Profiler (NanoString Technologies Inc., Seattle, WA, USA). The BC360 assay covers genes from 33 independent signatures, including the PAM50 signature (https://www.nanostring.com/, accessed date: 30 August 2019). By using multiplexed hybridization and digital readouts of fluorescent barcoded probes, the NanoString platform measures the relative abundance of each mRNA transcript of interest [20]. In a single reaction, the BC360^TM^ gene expression panel (including 758 gene-specific probe pairs of the BC360 targets, 18 housekeeping genes used for normalization, 6 exogenous positive control RNA targets, and 8 exogeneous negative control sequences) was hybridized in solution with 50–250 ng total RNA overnight (24 h) at 65 °C. The samples were processed using the NanoString nCounter^®^ SPRINT platform in accordance with the instructions and kits provided by NanoString Technologies (https://www.nanostring.com/, accessed date: 30 August 2019).

### 2.4. Molecular Subtype Classification

Reporter code count files containing the counts of each target mRNA molecule for every sample were sent to NanoString Technologies Inc. (Seattle, WA, USA). The raw data underwent a quality control check to ensure that PAM50 subtypes could be calculated. Only sample data that passed predefined quality control thresholds for the housekeeping gene geomean and the housekeeping gene signal-to-noise ratio were included in the PAM50 analysis. PAM50 breast cancer intrinsic subtype calling was conducted using a proprietary algorithm based on the 50-gene expression signature detailed by Parker et al. [21], ultimately classifying the samples into the four following subtypes: luminal A, luminal B, HER2-E, and basal-like. For each sample, the Pearson’s correlation coefficient for each of the four PAM50 centroids was calculated, and the subtype of the sample was assigned to the subtype of the centroid with the highest correlation.

### 2.5. Statistical Analysis

The associations between the PAM50 breast cancer intrinsic subtype and different patient and tumor characteristics were analyzed with Pearson’s chi-square test or Fisher’s exact test if the lowest expected count in a contingency table cell was <5. The exact McNemar test was used for comparisons of dichotomized PAM50 breast cancer intrinsic subtype status at two locations. When comparing all four subtypes at two locations, an exact test of symmetry was used. The null hypothesis of symmetry corresponded to a balanced subtype shift, and the alternative hypothesis corresponded to skewness in the shifts. Sankey diagrams were used to depict subtype shifts between the different locations.

The endpoints of the original trial were used to evaluate the prognostic value of the PAM50 breast cancer intrinsic subtype classification at each tumor site and PAM50 breast cancer intrinsic subtype shift versus no shift. The primary endpoint was the progression-free survival (PFS), and the secondary endpoint was the overall survival (OS) [19]. The progression-free survival was calculated from when the blood sample was collected on the day of inclusion in the study until the day that progression was diagnosed by radiology and/or clinical assessment. The follow-up time was censored at the date of the last follow-up for patients who did not reach the outcome event.

To evaluate the association between subtype shift and prognosis, Kaplan–Meier estimates and log-rank tests were used. Hazard ratios (HRs) were calculated using Cox regression. Multivariable Cox models were used to evaluate whether the PAM50-related variables added prognostic value to the prognostic variables suggested by Bidard et al. [22]. The variables included in the multivariable models were the following: age at diagnosis of metastatic breast cancer (<65 versus ≥65 years), ECOG performance status (0 versus 1 versus 2), Nottingham histologic grade (NHG) of PT (I versus II versus III), metastasis-free interval (MFI) (0 versus >0–3 versus >3 years), number of metastatic sites (<3 versus ≥3), site of metastasis (visceral versus non-visceral), and CTC status (<5 versus ≥5 CTC). The proportional hazards assumptions for the Cox models were checked graphically.

All presented statistical tests were two-sided. The *p*-values, which should be interpreted as the level of evidence against the null hypothesis, were not adjusted for multiple testing. Following the work of Benjamin et al. [23], the term ‘suggestive evidence’ was used for *p*-values in the range of 0.005–0.05, and the term ‘significant evidence’ was used for *p*-values below 0.005. Statistical calculations were conducted using IBM SPSS Statistics (version 24.0 and 26.0, IBM, Armonk, NY, USA) and STATA (version 16.1, StataCorp., College Station, TX, USA). The results are presented in accordance with the Reporting Recommendations for Tumor Marker Prognostic Studies (REMARK) where applicable [24,25].

## 3. Results

### 3.1. Patients and Tumor Characteristics

The original study included patients with newly diagnosed MBC who were previously untreated for metastatic disease. Of the 156 patients included in the original observational study, 140 (90%) were included in the final PAM50 analysis (Figure 1). The patient demographics, disease characteristics, and prior therapy for the original cohort and the included versus excluded subgroups are presented in Appendix A.

Overall, the characteristics of the 140 patients included in the present study were representative of the original cohort. However, non-assessable patients did tend to present with smaller PTs, less lymph node involvement, and fewer metastatic sites. The median follow-up time from inclusion in the study to progression was 49 months (range of 27–93 months) for patients alive at their last medical visit before database lock (April 2019). At this time point, 7% of the patients were progression-free and 29% were still alive. Twenty-nine patients presented with de novo MBC at the time of inclusion in the study, whereas 111 patients were diagnosed with distant recurrence.

### 3.2. PAM50 Breast Cancer Intrinsic Subtype Distribution and Association with Clinicopathologic Characteristics

The PAM50 breast cancer intrinsic subtypes could be assigned to 123 PTs, 68 LNMs, and 74 DMs, of which 33 patients had matched samples from all three locations (Figure 1 and Table 1). The characteristics of the PTs of the patients (*n* = 123) with the assignment of a PAM50 intrinsic molecular subtype were in line with those PTs included in the original study and the initial PAM50 cohort (Appendix A). The available DM biopsies were obtained from the following sites: bone (*n* = 43), liver (*n* = 17), skin (*n* = 3), central nervous system (CNS; *n* = 2), lung (*n* = 2), pleura (*n* = 2), stomach (*n* = 2), ovary (*n* = 1), bladder (*n* = 1), and cervix (*n* = 1). The distribution of the PAM50 breast cancer intrinsic subtypes determined for PTs and metastases (LNMs and DMs) is shown in Figure 1.

Overall, the most frequent PT subtypes were luminal A (38%, 47/123) and luminal B (37%, 45/123), followed by basal-like (13%, 16/123) and HER2-E (12%, 15/123). Regarding the subtype assignment for LNMs, luminal B was the most common subtype (54%, 37/68), followed by luminal A (26%, 18/68). Most DMs were also classified as luminal B (55%, 41/74), while the second most common DM subtype was HER2-E (26%, 19/74). Only 9% (7/74) of DMs were classified as luminal A.

Patient and tumor characteristics according to the PAM50 intrinsic subtypes in PTs, LNMs, and DMs are summarized in Table 1 and Appendix A, respectively. Evidence for an association with the PT PAM50 subtype was seen for the nodal status (*p* = 0.01), MFI (*p* = 0.02), NHG, estrogen receptor (ER), progesterone receptor (PR), and HER2-status, as well as for the clinically defined immunohistochemistry (IHC) molecular subtype (all *p*-values < 0.001).

Patients with the luminal A PT subtype were more frequently observed to have a longer MFI (76%, >3 years) compared to the other subtypes. Conversely, patients with a basal-like subtype PT generally had a shorter MFI (44% ≥0–3 years) (Table 1). Patients with de novo metastatic disease were found in all PAM50 intrinsic subtypes but most commonly presented with the luminal B subtype (42%, 11/26; Table 1).

### 3.3. Concordance between PAM50 Breast Cancer Intrinsic Subtypes and Clinically Defined ER and HER2 Expression Status

Excellent agreement was found between the PT ER-positive IHC status and luminal subtypes (A and B) in PTs (98% for both, 43/44; Table 1 and Appendix A). When comparing the HER2-positive status in PTs and the HER2-E intrinsic subtype in PTs, half of the tumor samples categorized as HER2-E intrinsic subtypes had HER2-positive status in the clinical pathological analyses of PTs (50%, 7/14), whereas four were ER-positive/HER2-negative, and three were triple-negative (TNBC). In line with these findings, HER2-positive tumors were categorized as being of the luminal B molecular subtype in 36% (5/14) of PTs.

Comparing intrinsic subtypes and the clinically assessed biomarker status of DMs revealed an excellent agreement (7/7 versus 35/35, i.e., 100%) between DMs with ER-positive IHC status and the luminal A and B subtypes (Appendix A). Comparing DMs and HER2-positive status with the intrinsic subtype showed that only 33% (6/18) of HER2-E DMs had a clinically defined HER2-positive tumor. Regarding all HER2-positive DMs, 75% (6/8) were categorized as being of the HER2-E subtype, whereas one was categorized as luminal B and one as basal-like.

### 3.4. PAM50 Breast Cancer Intrinsic Subtype Status across Tumor Progression Stages in Matched Samples

The PAM50 breast cancer intrinsic subtype distribution of matched samples is displayed in Figure 1, and changes in the matched tumor pairs and triplets are presented in Figure 2A–D and in Appendix A. Paired data between PTs and LNMs were available for a total of 59 cases for the PAM50 intrinsic subtype (Figure 2A) and for a total of 61 cases between PTs and DMs (Figure 2B).

In PTs with matching LNM samples (*n* = 59), 51% (30/59) were classified as luminal A, whereas only 29% (17/59) of matched LNMs were assigned the luminal A subtype. In paired samples from PTs and DMs (*n* = 61), 49% (30/61) of PTs were classified as luminal A, whereas only 11% (7/61) of matched DMs were luminal A. In samples with matched PTs, LNMs, and DMs (*n* = 33), 61% (20/33) of PTs were luminal A, but only 21% (7/33) of LNMs and 9% (3/33) of DMs were classified the same (Figure 1).

#### 3.4.1. Primary Tumors versus Matched Samples of Lymph Node Metastases

In matched samples of PTs and LNMs (*n* = 59), PAM50 breast cancer intrinsic subtype shifts were seen in 42% (25/59) of cases across all molecular subtypes (Figure 2A, Appendix A), and this finding was shown to be significantly skewed using the exact test of symmetry (*p* = 0.002). Shifts were seen in both directions; however, the most common shift was from the luminal A subtype to the luminal B subtype. In exact McNemar tests, a shift in subtype was seen in 36% (21/59) of cases considering luminal A versus not, where 81% (17/21) of case switched from the luminal A subtype to the luminal B subtype (*p* = 0.007).

#### 3.4.2. Primary Tumors versus Matched Samples of Distant Metastases

In matched samples of PTs and DMs (*n* = 61), shifts in subtype were seen in 51% (31/61) of cases (Figure 2B, Appendix A), and the shifts were shown to be significantly skewed by the exact test of symmetry (*p* < 0.001). The most common shift (*n* = 25) was from the luminal A PT subtype, with 16 cases shifting to luminal B, seven cases shifting to HER2E, and two cases shifting to the basal DM (McNemar test, *p* < 0.001).

#### 3.4.3. Lymph Node Metastases versus Matched Samples of Distant Metastases

Matched samples of LNMs and DMs were available for a total of 38 cases for the PAM50 subtype, and shifts in subtype were noted in 42% (16/38) of cases (Figure 2C, Appendix A). These shifts were found to be significantly skewed (*p* = 0.004). Between LNMs and DMs, the luminal B subtype changed in 40% (10/25) of cases; of these, shifts to the HER2-E or basal-like subtype were seen in 90% (9/10) of cases (Appendix A). For these 10 pairs with the luminal B subtype shift, the DMs were located at different sites (four in bone, two in the liver, one in the skin, one in the CNS, one in the ovaries, and one in the urinary bladder). Five of seven LNMs with the luminal A subtype switched to the luminal B subtype in DMs.

#### 3.4.4. Primary Tumors versus Matched Samples of Lymph Node Metastases and Distant Metastases

Thirty-three patients had matched samples from all three sites (PT, LNM, and DM). Overall, 64% (21/33) of the patients showed a shift in their PAM50 subtype, and this was toward a more unfavorable subtype for most of them (86%, 18/21) (Figure 2D and Appendix A).

### 3.5. Comparing PAM50 Breast Cancer Intrinsic Subtype Shifts from Luminal to Non-Luminal Subtypes between Tumor Sites

When comparing shifts from luminal (luminal A or luminal B) to non-luminal (HER2-E or basal-like) subtypes between tumor sites, differences were seen regarding shifts from PT to LNM versus shifts from PT to DM. A shift from the PT luminal subtype to the non-luminal subtype in LNMs was detected in only 6% (3/51) of patients (Figure 2A, Appendix A) compared to 23% (12/52) between PTs and DMs (Figure 2B, Appendix A). Between PTs and DMs, 40% (12/30) of shifts involved a change from the luminal subtype to the non-luminal subtype (exact McNemar test *p* < 0.001). Of these, 10/12 shifted to the HER2-E DM subtype (Figure 2B). A comparison between LNMs and DMs showed a shift from luminal to non-luminal in 24% (9/38; exact McNemar test *p* = 0.004) of cases, in which six cases with the LNM luminal (B) subtypes switched to the HER2-E subtype in DMs and three shifted to the basal-like subtype.

### 3.6. Associations between PAM50 Breast Cancer Intrinsic Subtype Shifts and Adjuvant Therapy

For patients that received adjuvant chemotherapy versus those who did not, there was no significant difference between patients with a stable subtype and those with subtype shifts (Appendix A). When considering subtype shifts from PT to DM in patients that received adjuvant endocrine therapy, there was a shift in 61% (27/44) of cases compared with 24% (4/17) of patients who did not receive adjuvant endocrine therapy (*p* = 0.01; Appendix A).

### 3.7. PAM50 Breast Cancer Intrinsic Subtype Status and Outcome in Patients with Available Subtype Assignment in PT, LNM, and DM

The PAM50 breast cancer intrinsic subtype of the PTs (*n* = 123) was significantly associated with PFS (*p* = 0.001) and OS (*p* < 0.001) for metastatic disease (Figure 3A,B). Patients assigned the PT basal-like subtype had worse outcomes than those with the other subtypes (median PFS of 5 months versus 11–18 months, respectively, and median OS of 11 months versus 35–45 months, respectively). When analyzing the LNM intrinsic subtype (*n* = 68) and the correlation with survival, there was suggestive evidence for an association with PFS (*p* = 0.027) but not OS (*p* = 0.058; Figure 3C,D), and DM intrinsic subtypes (*n* = 74) had no significant associations with PFS or OS (*p* = 0.096 and *p* = 0.907, respectively; Figure 3E,F).

In Cox proportional hazards models, the PAM50 subtype of the PTs remained an independent prognostic factor for both PFS (*p* = 0.007) and OS (*p* = 0.001) in multivariable models adjusted for clinically relevant prognostic factors (Table 2). Regarding the LNM intrinsic subtype, there were no significant correlations with PFS or OS in the adjusted Cox regression models. However, in the adjusted analyses, a significant difference in PFS was observed when the PAM50 intrinsic subtypes of the DMs were considered (*p* = 0.002; 3-degree-of-freedom (df) overall test), and there was suggestive evidence for a correlation with OS (3-df *p* = 0.05; Table 2).

### 3.8. PAM50 Breast Cancer Intrinsic Subtype Shifts and Outcome in Patients with Paired Samples

In comparing tumors with shifts between luminal and non-luminal subtypes (luminal to non-luminal and non-luminal to luminal) versus tumors with a stable subtype (luminal stable and non-luminal stable), suggestive evidence of an association between these subgroups and PFS was seen between matched PT and LNM samples (*n* = 59) (*p* = 0.01, log-rank test; Appendix A) and between matched PT and DM samples (*n* = 61) (*p* = 0.03, log-rank test; Appendix A).

Patients with a shift from a luminal PT subtype to a non-luminal DM subtype (*n* = 12) had an inferior PFS compared with patients with a stable luminal subtype (*n* = 40, log-rank; HR: 2.31; 95% CI: 1.14–4.68, *p* = 0.02; Figure 4A). The same tendency was seen regarding the OS when comparing patients with a stable luminal subtype to patients with a shift from a luminal-like subtype in the PT to a non-luminal subtype in the DM (log-rank; HR: 1.86; CI: 0.87–4.0, *p* = 0.11; Figure 4B).

## 4. Discussion

To test the hypothesis that molecular subtypes are inconsistent during tumor progression in MBC patients, we investigated PAM50 intrinsic breast cancer subtypes in samples of PTs, synchronous LNMs, and DMs. We found strong evidence that PAM50 breast cancer intrinsic subtypes are unstable during tumor progression. Breast cancer intrinsic subtype shifts of 42–51% were detected between tumor progression sites, and the majority of subtype shifts were toward more aggressive subtypes in metastases. The observed shift towards an unfavorable subtype from PT to DM confirms previous data regarding bio-marker status, whereas the significant shift toward a more aggressive molecular subtype in synchronous LNMs has, to our knowledge, not been previously described [9,10,11]. The pronounced conversion from a luminal PT subtype to a non-luminal DM subtype was associated with a worse prognosis, underscoring the relevance of the re-evaluation of the molecular subtypes in metastatic biopsies.

Previous studies have compared breast cancer molecular subtypes between tumor progression sites by means of IHC-based St Gallen classification [9,10,11] or using PAM50 gene expression signatures [9,12,13,14,15,16]. Similar to our results, these publications found that while a large portion of paired (matched) samples maintained the same molecular subtype, shifts also occurred over time in multiple cases, with a trend of primarily changing from the luminal A subtype to more unfavorable subtypes.

The reason why these shifts in molecular subtype inherence occur between tumor sites remains speculative. However, in line with our and previous results, a recently published mathematical cancer progression model developed by Chen and colleagues that included data from matched primary and metastatic tumor samples supports a directional linear evolution through luminal subtypes to increasingly aggressive subtypes [15]. We also detected subtype shifts in the other direction, i.e., the luminal B and HER2-E subtypes shifting to the luminal A subtype, emphasizing the possible plasticity and complexity of tumor progression and cancer evolution. Other explanations could be intratumor heterogeneity and the fact that multiple breast cancer subtypes co-exist within the same tumor [26,27].

This is supported the work of by Lluch et al. [16], who found an association between clinical subtype changes and dynamic clonal remodeling in breast cancer metastases. In addition, a shift in molecular status may mirror changes following adjuvant therapy [28]. We did see an association between PAM50 subtype shifts and adjuvant endocrine treatment, indicating the possibility that tumor cells that survive adjuvant therapy are likely to be endocrine-resistant clones that are more able to metastasize (e.g., non-luminal subtypes). The determined association reflects the fact that cases where there is a shift in subtype between tumor sites are most often classified as luminal A PT subtypes. Additionally, the significant shift in the PAM50 subtype between PTs and synchronous LNMs suggests that subtype shifts could be independent of treatment effects. It is also plausible that several of the abovementioned factors could occur concurrently.

It is intriguing that the luminal A subtype appeared to be the most unstable subtype throughout tumor progression given that this is the most frequently observed subtype in early breast cancer and is associated with superior prognosis. In addition, patients with this subtype are less heavily treated compared to patients with other subtypes [29,30,31]. However, the cohort in our study represented patients with metastatic disease, where most cases had node positive disease at the initial diagnosis: hence, our study is not representative of all early breast cancer cases with the luminal A subtype.

In this context, it is important to note that the luminal A subtype has been associated with the highest diversity in clinical outcomes as well as with the highest inter- and even intratumor heterogeneity [26,32,33]. A recent study reported worse outcomes in patients with a mixed subclonal luminal A subtype compared to patients with a more ‘pure’ luminal A subtype [26]. Taken together with our results, these findings highlight the need to stratify the luminal A subtype even further to be able to optimize treatment for these patients.

The shifts in tumor inherence observed in the present study suggested that the intrinsic molecular subtyping of metastatic disease provides additional predictive information that can be used to determine prognosis and therapy. The shift from luminal to non-luminal subtypes observed between PTs and synchronous LNMs (Figure 2A) indicated that this change might be an important consideration when selecting adjuvant treatment. Importantly, patients in which there is conversion from a luminal subtype in the PT to the HER2-E subtype in DMs may benefit from anti-HER2 treatment, as was previously suggested for clinically HER2-negative MBC [34], again highlighting the potential importance of the molecular characterization of metastatic disease. In the present study, we found a suggestive significant difference in PFS between the subgroups of patients with a stable luminal subtype and those with a shift from a luminal subtype in the PT to a non-luminal subtype in DM.

Many studies have focused on the importance of PAM50 subtyping in the early stages of breast cancer, whereas few studies have investigated the prognostic value of PAM50 subtyping in the metastatic setting and instead mostly evaluated prognosis in relation to the PAM50 subtype of the primary tumor [9,12,34,35,36]. The present study has demonstrated the added prognostic value of determining the PAM50 breast cancer intrinsic PT subtype in patients with MBC [9,13,35,36]. In line with our findings, Cejalvo et al. found that PAM50 intrinsic PT subtyping had a much greater impact on the overall survival than the subtyping of DMs [13].

The presence of luminal B and HER2-E subtypes during tumor progression in the present study corresponds to the findings of Cejalvo et al. and might represent tumors that lose their estrogen dependence while acquiring a worse phenotype [13]. Interestingly, a recently proposed prognostic model for ER-positive/HER2-negative MBC by Prat et al. combined clinical variables and PAM50 subtyping that was adjusted for the type of biopsy (primary versus metastasis), and it showed a high discriminatory property for predicting prognosis. Their data also supported that PAM50 subtyping adds prognostic information in MBC [37]. The prognostic value of subtyping LNMs according to the St Gallen molecular subtype has previously been reported [9,11].

However, we were not able to demonstrate a similar strong prognostic value for subtyping LNMs using the PAM50 subtype classifier. Tobin et al. [38,39] evaluated PAM50 using fine-needle aspirates from synchronous LNMs and asynchronous DMs in relation to post-relapse survival and found that the PAM50 signature was an independent prognostic factor when considering LNMs. One important difference between the present study and the study presented by Tobin et al. is that our study included previously systemically untreated patients with distant metastasis and not those with locoregional disease.

The present study had several strengths. Primarily, it was one of the largest studies to evaluate the PAM50 intrinsic subtype classification of samples, including distant metastases, from a cohort of newly diagnosed MBC patients who were previously untreated for metastatic disease and who were included in a prospective observational study. All multivariable models of PFS and OS were adjusted for relevant prognostic factors used in clinical practice, as suggested by Bidard et al. [22]. We included samples from 90% of the participants in the original trial, applied the validated PAM50 intrinsic subtyping assay in a highly standardized manner, and prospectively defined hypotheses and a statistical analysis plan.

However, the study also had certain limitations. The study had a prospective–retrospective design, where patients had received various systemic treatments, and some of the subgroup analyses had limited statistical power due to the small group size. Even though all included patients had metastatic disease, distant metastatic biopsies were not always available as FFPE samples because the diagnosis of some patients had only been confirmed using computed tomography or cytology examinations. In addition, several patients had more than one synchronous LNM and/or DM; however, we only evaluated one LNM and the first DM in the present analysis. This could have contributed to sampling error and selection bias because the evaluated metastasis might not represent the full metastatic burden in the patient. Finally, studying the dynamic process of tumor progression in momentary ‘snapshots’ of tumor biopsies at different stages only reflects part of the picture.

## 5. Conclusions

The results of this study confirmed PAM50 molecular subtype shifts during tumor progression. The directional tumor evolution from luminal subtypes to more unfavorable subtypes was shown, indicating the need for the proper molecular characterization of metastatic disease to improve the personalization of systemic therapy. Importantly, the prognosis in patients displaying shifts from luminal subtypes in PTs to basal-like and HER2-enriched inherence in DMs was inferior to that of patients with a stable luminal subtype. A shift in the breast cancer molecular subtype during tumor progression adds predictive information that is clinically relevant for the determination of prognosis and treatment.

## Figures and Tables

**Figure 1 cancers-13-01592-f001:**
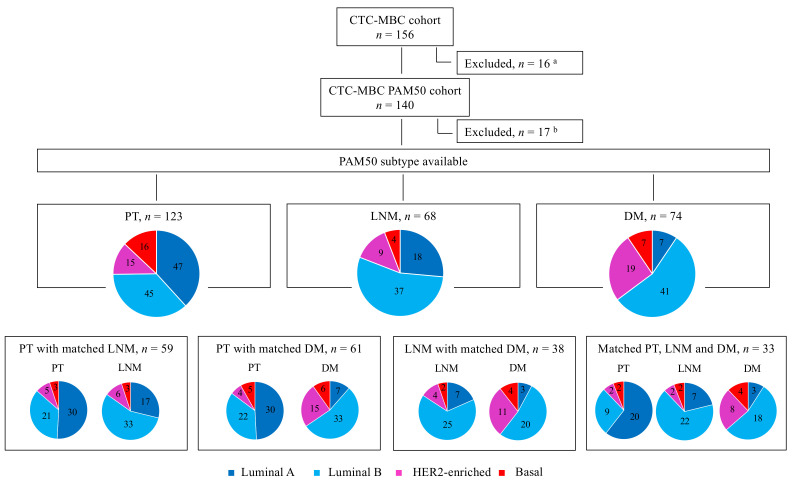
Flow chart of the study population and PAM50 breast cancer intrinsic subtype distribution between tumor sites. ^a^ Patients were excluded from the study population due to any of the following reasons: sample being unavailable, no tumor tissue in the available sample, only locoregional sample available, or unsuccessful RNA extraction. ^b^ Patients were excluded due to unsuccessful NanoString Breast Cancer 360 (BC360) assay or because PAM50 subtypes could not be assigned. Abbreviations: CTC: circulating tumor cell; DM: distant metastasis; HER2: human epidermal growth factor receptor 2; LNM: lymph node metastasis; MBC: metastatic breast cancer; PT: primary tumor.

**Figure 2 cancers-13-01592-f002:**
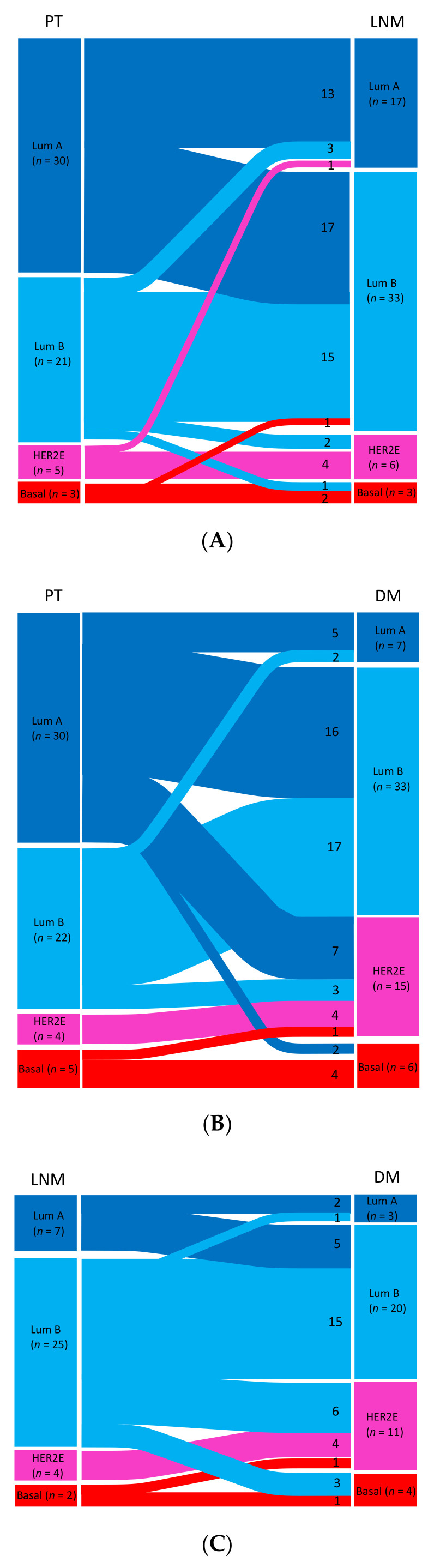
(**A**–**D**): Sankey diagrams showing the PAM50 breast cancer intrinsic subtype shifts of matched samples between (**A**) the primary tumor (PT) and lymph node metastasis (LNM) (*n* = 59); (**B**) the PT and distant metastasis (DM) (*n* = 61); (**C**) LNM and DM (*n* = 38); and (**D**) the PT, LNM, and DM (*n* = 33). Abbreviations: HER2-E: human epidermal growth factor receptor 2-enriched; Lum A/B: luminal A/B.

**Figure 3 cancers-13-01592-f003:**
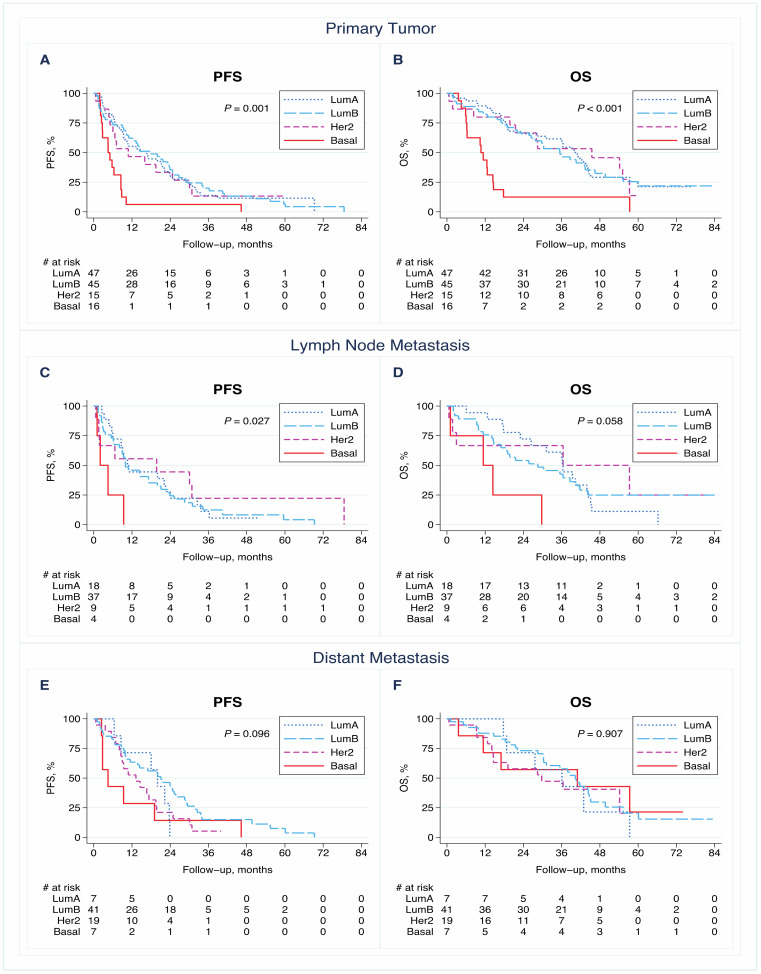
Kaplan–Meier survival curves for progression-free survival (PFS) and overall survival (OS) according to PAM50 breast cancer intrinsic subtype in the primary tumor (**A**,**B**), lymph node metastasis (**C**,**D**), and distant metastasis (**E**,**F**). *p*-values were calculated using the log-rank test. Abbreviations: HER2: human epidermal growth factor receptor 2; Lum A/B: luminal A/B.

**Figure 4 cancers-13-01592-f004:**
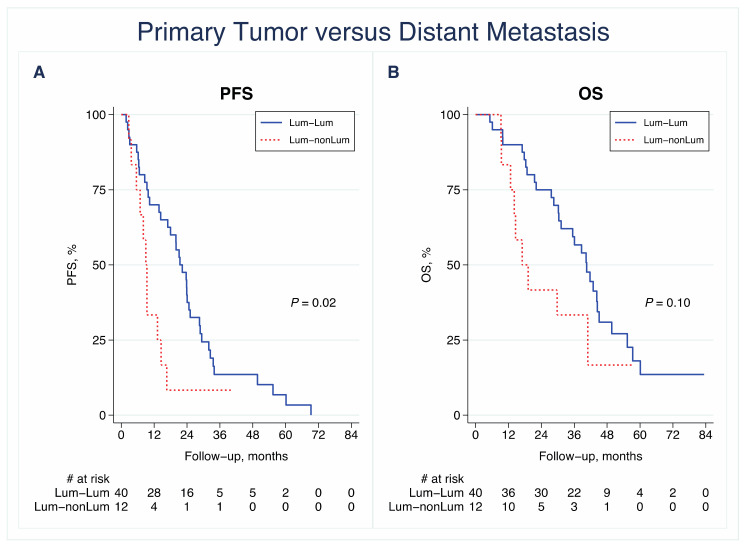
Kaplan–Meier survival curves for (**A**) progression-free survival (PFS) and (**B**) overall survival (OS) in relation to subgroups of patients with shifts between luminal (luminal A or B (Lum)) and non-luminal (HER2-E or basal-like (nonLum)) subtypes from the primary tumor to distant metastasis. *p*-values were calculated using the log-rank test.

**Table 1 cancers-13-01592-t001:** Patient and tumor characteristics for primary tumors in the PAM50 cohort and subdivided into patients with subtype available and the different breast cancer intrinsic subtypes.

Characteristics	PAM50 Subtype Available *n* (%)	Lum A *n* (%)	Lum B *n* (%)	HER2-E *n* (%)	Basal-Like *n* (%)	*p* ^a^
**All**	123	47	45	15	16	
**Age at MBC diagnosis (years)**						0.20 ^d^
<65	63 (51)	22 (47)	20 (44)	10 (67)	11 (69)	
≥65	60 (49)	25 (53)	25 (56)	5 (33)	5 (31)	
**ECOG (BL)**						0.10 ^e^
0	70 (58)	34 (75)	21 (48)	7 (47)	8 (50)	
1	31 (26)	8 (18)	14 (32)	5 (33)	4 (25)	
2	19 (16)	3 (7)	9 (20)	3 (20)	4 (25)	
Missing	3	2	1	0	0	
**Size (PT)**						0.77 ^e^
T1	38 (33)	15 (33)	13 (29)	3 (27)	7 (43)	
T2	43 (37)	17 (38)	18 (40)	5 (46)	3 (19)	
T3	19 (16)	9 (20)	6 (13)	1 (9)	3 (19)	
T4	17 (14)	4 (9)	8 (18)	2 (18)	3 (19)	
Missing	6	2	0	4	0	
**Nodal status** ^b^						0.01 ^e^
Negative	31 (29)	7 (17)	11 (27)	4 (36)	9 (64)	
Positive	77 (71)	35 (83)	30 (73)	7 (64)	5 (36)	
Missing	15	5	4	4	2	
**NHG (PT)**						<0.001 ^e^
I	8 (8)	7 (18)	1 (3)	0	0	
II	50 (51)	28 (72)	15 (39)	6 (60)	1 (9)	
III	40 (41)	4 (10)	22 (58)	4 (40)	10 (91)	
Missing	25	8	7	5	5	
**ER (PT)**						<0.001 ^e^
Negative	20 (17)	1 (2)	1 (2)	6 (40)	12 (75)	
Positive	99 (83)	43 (98)	43 (98)	9 (60)	4 (25)	
Missing	4	3	1	0	0	
**PR (PT)**						<0.001 ^e^
Negative	37 (32)	4 (10)	9 (21)	10 (67)	14 (88)	
Positive	80 (68)	38 (90)	35 (79)	5 (33)	2 (12)	
Missing	6	5	1	0	0	
**HER2 (PT)**						<0.001 ^e^
Negative	89 (86)	32 (94)	36 (88)	7 (50)	14 (100)	
Positive	14 (14)	2 (6)	5 (12)	7 (50)	0	
Missing	20	13	4	1	2	
**BC subtype (PT)**						<0.001 ^e^
ER+ HER2−	75 (73)	32 (94)	35 (88)	4 (29)	4 (29)	
HER2+ (ER+/ER−)	13 (13)	2 (6)	4 (10)	7 (50)	0	
ER− HER2−	14 (14)	0	1 (2)	3 (21)	10 (71)	
Missing	21	13	5	1	2	
**MFI (years)**						0.02 ^e^
0 ^c^	26 (21)	7 (15)	11 (24)	5 (33)	3 (19)	
>0–3	23 (19)	4 (9)	8 (18)	4 (27)	7 (44)	
>3	74 (60)	36 (76)	26 (58)	6 (40)	6 (37)	
**Number of metastatic sites**						0.70 ^d^
1–2	82 (67)	33 (70)	30 (67)	8 (53)	11 (69)	
≥3	41 (33)	14 (30)	15 (33)	7 (47)	5 (31)	
**Metastatic site**						0.26 ^d^
Non-visceral	51 (42)	23 (49)	19 (42)	3 (20)	6 (38)	
Visceral	72 (58)	24 (51)	26 (58)	12 (80)	10 (62)	
**Number of CTCs (BL)**						0.32 ^d^
<5	57 (46)	25 (53)	16 (36)	7 (47)	9 (56)	
≥5	66 (54)	22 (47)	29 (64)	8 (53)	7 (44)	
Missing	0	0	0	0	0	

Abbreviations: BL: baseline; BC: breast cancer; CTCs: circulating tumor cells; ECOG: Eastern Cooperative Oncology Group; ER: estrogen receptor; HER2: human epidermal growth factor receptor 2; HER2-E: human epidermal growth factor receptor 2 enriched; Lum A/B: luminal A/B; MBC: metastatic breast cancer; MFI: metastasis-free interval; NHG: Nottingham Histological Grade; *n*: number; PT: primary tumor; PR: progesterone receptor. ^a^ Test of homogeneity across subtypes, missing value category excluded; ^b^ at primary diagnosis; ^c^ represents de novo MBC; ^d^
*p*-value from Pearson’s chi-square test; ^e^
*p*-value from Fisher’s exact test.

**Table 2 cancers-13-01592-t002:** Univariable and multivariable Cox regression models of PAM50 breast cancer intrinsic subtypes in primary tumors, lymph node metastases, and distant metastases.

	Univariable PFS			Multivariable PFS ^a^			Univariable OS			Multivariable OS ^a^		
	*n*	HR (95% CI)	*p*	*n*	HR (95% CI)	*p*	*n*	HR (95% CI)	*p*	*n*	HR (95% CI)	*p*
**PT**	123			89			123			69		
PAM50 subtype			0.002 ^b^			0.007 ^b^			0.001 ^b^			0.001 ^b^
Lum A	47	1.00			1.00			1.00			1.00	
Lum B	45	0.94 (0.61–1.4)	0.79		0.93 (0.54–1.6)	0.81		1.1 (0.66–1.8)	0.77		0.92 (0.48–1.8)	0.81
HER2-E	15	1.1 (0.58–2.0)	0.82		0.88 (0.38–2.0)	0.77		1.0 (0.52–2.1)	0.92		0.77 (0.31–1.9)	0.57
Basal-like	16	2.8 (1.6–5.1)	0.001		3.8 (1.5–9.3)	0.004		3.4 (1.8–6.3)	<0.001		4.6 (1.8–11.3)	0.001
**LNM**	68			55			68			44		
PAM50 subtype			0.05 ^b^			0.28 ^b^			0.09 ^b^			0.17 ^b^
Lum A	18	1.00			1.00			1.00			1.00	
Lum B	37	1.1 (0.6–1.9)	0.84		0.88 (0.41–1.9)	0.76		0.93 (0.5–1.7)	0.81		1.2 (0.53–2.8)	0.63
HER2-E	9	0.73 (0.3–1.8)	0.48		0.41 (0.13–1.3)	0.12		0.63 (0.23–1.7)	0.37		0.45 (0.13–1.6)	0.23
Basal-like	4	4.2 (1.4–13)	0.01		1.5 (0.37–6.4)	0.55		3.4 (1.1–10.3)	0.04		2.7 (0.64–11.6)	0.18
**DM**	74			58			74			42		
PAM50 subtype			0.11 ^b^			0.002 ^b^			0.91 ^b^			0.05 ^b^
Lum A	7	1.00			1.00			1.00			1.00	
Lum B	41	0.60 (0.26–1.4)	0.22		0.26 (0.08–0.80)	0.02		0.77 (0.32–1.9)	0.56		0.46 (0.13–1.6)	0.22
HER2-E	19	1.0 (0.42–2.5)	0.97		0.64 (0.20–2.1)	0.46		0.88 (0.33–2.4)	0.81		1.6 (0.40–6.8)	0.49
Basal-like	7	1.4 (0.46–3.9)	0.58		2.4 (0.54–10.3)	0.26		0.69 (0.21–2.3)	0.55		1.3 (0.24–6.4)	0.79

Abbreviations: CI: confidence interval; DM: distant metastasis; ECOG: Eastern Cooperative Oncology Group; HR: hazard ratio; HER2-E: human epidermal growth factor receptor 2 enriched; LNM: lymph node metastasis; MFI: metastasis-free interval; NHG: Nottingham Histological Grade; *n*: number; OS: overall survival; PT: primary tumor; PFS: progression-free survival. ^a^ Adjusted for age at diagnosis of metastatic breast cancer (<65 versus ≥65 years), ECOG performance status (0 versus 1 versus 2), NHG of PT (I versus II versus III), MFI (0 versus >0–3 versus >3 years), number of metastatic sites (<3 versus ≥3), site of metastasis (visceral versus non-visceral), and circulating tumor cell (CTC) status (<5 versus ≥5 CTCs); ^b^ 3-degree-of-freedom overall test.

## Data Availability

The datasets used and/or analyzed during the current study are available from the corresponding author upon reasonable request.

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
