# Peer review of "PAM50 Intrinsic Subtype Profiles in Primary and Metastatic Breast Cancer Show a Significant Shift toward More Aggressive Subtypes with Prognostic Implications"

_cancers, 2021, doi:10.3390/cancers13071592_

Round 1
Reviewer 1 Report
The study is aimed to investigate the changes of breast cancer subtypes (PAM50 intrinsic subtypes) throughout the progression of the disease in a cohort of 140 patients with metastatic breast cancer, and whether those changes are correlated with a poorer prognosis. To do so, RNA was extracted from formalin-fixed paraffin-embedded tumor tissue from matching primary tumors, lymph-node metastases and distant metastases, and gene expression profiles were obtained with a NanoString panel (BC360) that includes PAM50 signature. All samples were validated for concordance between PAM50 intrinsic subtype and clinically defined molecular subtype by immunohistochemistry. A straightforward approach was followed to select the cases, as well as a robust statistical analysis of the results. In summary, this study found a shift of the PAM50 molecular subtypes during tumor progression, from luminal to a more aggressive subtype and therefore, to a disease with poorer prognosis.
This is an elegant paper; English language is correct, is a beautifully written manuscript, methodology and experimental design are correct, a robust statistical approach was made, and the results are very clearly presented, especially Sankey diagrams resulted very helpful to follow the shifts in tumor subtypes throughout tumor progression.
A couple of minor comments could be done:
- Table 2: consider a horizontal design in the manuscript.
- Figure 4: “A” and “B” are missing in the figure, whereas they are mentioned in the text as Figure 4A and Figure 4B. Please, fix.
Author Response
- We agree that a horizontal design of Table 2 is preferrable and will forward this query to the publisher.
- Thank you for pointing this out. We have now clarified the Figure 4A and the Figure 4B in the legend.
Reviewer 2 Report
the paper is clear and well writtent
the authors present a retro/prospective series of breast cancer patients, with a matched PAM 50 analysis of the primary tumor, with when abailable regional lymph nodes and/or metastatic tissue. The analysis is well conducted and methods are sound.
this issue has been partly addressed by some retrospective series. The originality here is the partly prospective part, and the combined primary tumor/lymph nodes/metastasis analysis, which brings interesting new and conforting data in the field
the text is clear and easy to read and the conclusions consistent with the evidence and arguments presented
authors address the main question posed
Author Response
Thank you for your comments.
Reviewer 3 Report
The manuscript by Jørgensen and Larsson et al., presents an extensive analysis of breast cancer PAM50 Intrinsic Subtype by expression profiling of primary and metastatic tumors. The research advances the understanding of the breast cancer molecular subtypes and can guide clinical strategies. Overall, the study is presented well and is suitable for publication. I recommend the following suggestions for improving the manuscript.
- Authors should consider comparing the results from the PAM50 intrinsic subtype of primary and metastatic tumors with other publicly available data sets, such as from TCGA. CbioPortal (https://www.cbioportal.org/) has data from multiple studies for Breast Cancer and includes gene expression based on RNA-seq. It would be interesting to see whether the findings described in this paper, mainly derived FFPE tissues, can be extrapolated and observed in the RNA-seq data from other studies, which have used more bulk and fresh frozen tumor materials.
- Line 109-112. Because RNA was prepared from FFPE samples, authors should provide a brief description and information on quality RNA such as average RIN score numbers, etc.
- Figures 3 and 4: The authors should avoid using abbreviations for subtypes and tumor sites on the plots. For example, there is enough space to have ‘PT’ as a Primary tumor. It is easy to follow full names when space is available on plots.
Author Response
- The introduction now includes a clarifying sentence on type of samples used in previous publications on line 72-75.
“The assessments of change in tumor inherence are based on analysis of archival tissues and mainly present the surrogate subtypes [5, 9, 10, 17] although one of them has presented data on the intrinsic molecular subtypes [13].”
- Thank you for pointing this out to us. We have not found any comparable dataset using RNAseq subtype calling including matched samples from primary tumors, lymph node metastasis and recurrences by searching the literature or in the TCGA datasets, so we are at the moment not able to decipher this issue with the data at hand.
In the CbioPortal, the TCGA data was explored and in the TCGA, Firehose Legacy, only 7 of 1108 samples are from a metastatic biopsy. On the other hand, the distribution of the intrinsic molecular subtypes for primary tumors in our cohort was compared with the TCGA dataset in Breast Invasive Carcinoma (TCGA, PanCancer Atlas; n = 1083) and is presented below:
TCGA Tykjaer Jörgensen
Luminal A 46.0% Luminal A 38%
Luminal B 18.2% Luminal B 37%
HER2-E 7.2% HER2-E 12%
Basal-like 15.8% Basal-like 13%
Normal-like 3.3%
Not annotated 9.5%
If the higher proportions of the luminal B and HER2-enriched subtypes in our cohort reflects a true difference in the methods for subtype calling and use of fresh bulky tissue or rather the fact that our cohort represents patients experiencing a recurrence cannot be sorted out.
Your comment is worth considering for a future meta-analysis when more data is available from RNAseq based data from metastasis and can be compared with our data and similar datasets (for example Cejalvo et al.).
- In the flow-chart (Figure 1), the exclusion of patients due to failure of subtype calling by the BC360 is presented (17/140 patients were excluded because their samples did not meet the predefined NanoString criteria for RNA content).
We have now clarified this on line 116-120.
“A subset of samples was analyzed with the Bioanalyzer as part of the quality control (QC) of the RNA (mean RIN-value was 2.3 (1.3–2.8). NanoString used multiple QC cut-offs to determine if a sample was PASS, FAIL, or BORDERLINE, which indicated if the RNA quality was high enough to produce reliable subtypes and that the NanoString assay should perform as expected. Due to these post collection QC steps, NanoString did not need to assess RNA quality before the assay. Most samples (90%) in the study passed these predefined quality criteria set by NanoString and only 17 patients were excluded due to analytical failure (Figure 1).”
- The Figures 3 and 4 are now updated for abbreviations as requested.